# Biofilm Formation of *Listeria monocytogenes* and *Pseudomonas aeruginosa* in a Simulated Chicken Processing Environment

**DOI:** 10.3390/foods11131917

**Published:** 2022-06-28

**Authors:** Qingli Dong, Linjun Sun, Taisong Fang, Yuan Wang, Zhuosi Li, Xiang Wang, Mengjie Wu, Hongzhi Zhang

**Affiliations:** 1School of Health Science and Engineering, University of Shanghai for Science and Technology, Shanghai 200093, China; dongqingli@126.com (Q.D.); sn_stzh@163.com (L.S.); usstfts@163.com (T.F.); lizhuosi@usst.edu.cn (Z.L.); xiang.wang@usst.edu.cn (X.W.); wumengjie202102@163.com (M.W.); 2School of Food and Drug, Shanghai Zhongqiao Vocational and Technical University, Shanghai 201514, China; wy40714034@163.com; 3Shanghai Municipal Center for Disease Control and Prevention, Shanghai 200336, China

**Keywords:** *Listeria monocytogenes*, *Pseudomonas aeruginosa*, biofilm formation, chicken processing environment

## Abstract

This study aims to investigate the mono- and dual-species biofilm formation of *Listeria monocytogenes* and *Pseudomonas aeruginosa* incubated in different culture mediums, inoculum ratios, and incubation time. The planktonic cell population and motility were examined to understand the correlation with biofilm formation. The results showed that chicken juice significantly inhibited the biofilm formation of *L. monocytogenes* (*p* < 0.05). *Pseudomonas aeruginosa* was the dominant bacteria in the dual-species biofilm formation in the trypticase soy broth medium. The dynamic changes in biofilm formation were not consistent with the different culture conditions. The growth of planktonic *L. monocytogenes* and *P. aeruginosa* in the suspension was inconsistent with their growth in the biofilms. There was no significant correlation between motility and biofilm formation of *L. monocytogenes* and *P. aeruginosa*. Moreover, scanning electron microscopy (SEM) results revealed that the biofilm structure of *L. monocytogenes* was loose. At the same time, *P. aeruginosa* formed a relatively dense network in mono-species biofilms in an initial adhesion stage (24 h). SEM results also showed that *P. aeruginosa* was dominant in the dual-species biofilms. Overall, these results could provide a theoretical reference for preventing and controlling the biofilm formation of *L. monocytogenes* and *P. aeruginosa* in the food processing environment in the future.

## 1. Introduction

Biofilms are complex structured bacterial communities that irreversibly adhere to biotic or abiotic surfaces with extracellular polymeric substances (EPS) [1]. Biofilms are responsible for the prolonged persistence of the bacteria in the food environment [2]. Once formed, a biofilm can protect bacteria from environmental stresses, antibiotics, and disinfectants and is challenging to decimate. Compared with planktonic bacteria, biofilm formation can protect bacteria from extreme environments, antibiotics, and disinfectants [3]. Biofilms can adhere to the surface of equipment or foods, grow and produce EPS to attach to the food contact surface, and therefore become an irremediable source of contamination that may lead to major disease outbreaks and substantial economic losses [4,5].

Biofilm formation is affected by many factors, such as the type of bacteria and the interaction among bacteria, media, and cultivation time [6,7]. Several studies have shown a significant difference between the amount of biofilm formation in laboratory culture media and food substrates [8]; therefore, the amount of biofilm formation in the laboratory simulating food environment is not necessarily reliable, and extensive studies on natural foods are currently lacking. The complex microorganisms in the food environment can lead to biofilm formation with mono- and dual-species or even multispecies strains. The interaction between the two bacteria or multibacteria and their ratios affects the biofilm formation [9]. There is a lack of understanding of the interaction effect of the dual-species biofilm formation, and further research is required. In addition, biofilm formation is a dynamic process, and the difference in culturing time also determines the amount and structure of biofilms [10]. Moreover, the relationship between the bacterial population, motility, and biofilm formation is still unclear.

*Listeria monocytogenes* is a gram-positive foodborne pathogen that can lead to a rare but high-fatality-rate disease called listeriosis [11]. *Listeria monocytogenes* often can be found in raw meat, milk, and ready-to-eat salad. The European Food Safety Authority (EFSA) and the European Centre for Disease Prevention and Control (ECDC) reported that 2549 cases of foodborne diseases, including 229 cases of death, were caused by *L. monocytogenes* in 2018 [12]. According to the recent survey by the EFSA and ECDC, a total of 1876 confirmed cases of invasive listeriosis in humans were reported in the European Union member states, with 97.1% hospitalizations [13]. According to the US Centers for Disease Control and Prevention (CDC), nine reported infections comprised three cases of hospitalization and one death attributed to *L. monocytogenes* in a fully cooked chicken [14]. The average prevalence of *L. monocytogenes* in the Chinese food products in 28 provinces was 4.42%, with the highest prevalence of 8.91% in the meat and poultry products, followed by aquatic animals, Chinese salad, salad, rice, and flour products [15].

*Pseudomonas* spp. can quickly produce dense biofilms, which result in food spoilage [16]. *Pseudomonas aeruginosa* has been studied as a model bacterium for the research of biofilm formation in raw meats [17]. The biofilm formation of *L. monocytogenes* and *P. aeruginosa* has attracted worldwide attention because of their role as pathogens and spoilage bacteria in the food environment. However, the effect of different ratios of bacteria on dual biofilm formation is unknown [8]. Additionally, only a few studies have been reported describing biofilm formation between *L. monocytogenes* and *Pseudomonas* spp. [18,19,20], mainly focused on mimicking the actual food processing environment. Based on this, it is of practical significance to study the dynamic changes in *L. monocytogenes* and *P. aeruginosa* biofilm formation in different culture mediums and mixing ratios.

The study aimed to evaluate the dynamic changes in biofilm formation of *L. monocytogenes* and *P. aeruginosa* in different culture mediums and mixing ratios, to observe the ability of biofilm formation with time, to measure the population of planktonic cells growth and motility of *L. monocytogenes* and *P. aeruginosa* during the formation of the biofilm, and to correlate the relationship between two attributes and biofilm formation. Studying the formation of *L. monocytogenes* and *P. aeruginosa* in mono- and dual-species biofilm under different conditions can provide a theoretical reference for preventing and controlling biofilm formation in the natural food environment.

## 2. Materials and Methods

### 2.1. Bacterial Strains and Medium

*L. monocytogenes* EGD-e (serovar 1/2a) and *P. aeruginosa* (ATCC 27853) were purchased from the American Type Culture Collection (Manassas, VA, USA). *Listeria monocytogenes* EGD-e and *P. aeruginosa* were transferred into 10 mL trypticase soy-yeast extract broth (TSB-YE, Qingdao Hope Bio-Technology Co., Ltd., Qingdao, China) and trypticase soy broth (TSB, Qingdao Hope Bio-Technology Co., Ltd., Qingdao, China), followed by incubation at 37 °C for 16–18 h at 110 rpm. After two consecutive transfers, the cells were pelleted at 8000× *g* for 10 min and washed in saline solution three times. The final concentration was adjusted to 10^4^ or 10^5^ CFU/mL for further experiments.

### 2.2. Preparation of Chicken Juice and Stainless Steel Coupons

The chicken juice was prepared according to Pang et al. with slight modifications [6]. Raw chicken breasts were bought online (Fresh daily App, Shanghai, China), and 250 g of chicken breasts were placed in a sterile stomacher bag with 300 mL of sterile distilled water. The chicken breasts were homogenized for 4 min and centrifuged at 12,000× *g* for 15 min to remove chicken debris. The chicken juice was filtrated by 0.22 mm pore-size filters for sterilization. The sterilized chicken juice was stored at −20 °C for four weeks, and the juice was placed at 4 °C one night before the experiments.

Stainless steel coupons (304; 2 cm × 1 cm × 0.2 cm) were soaked in acetone (Qingdao Hope Bio-Technology Co., Ltd., Qingdao, China) for 3 h and cleaned with kitchen towels. After immersing in ethanol (Qingdao Hope Bio-Technology Co., Ltd., Qingdao, China) overnight, the coupons were rinsed with distilled water and air-dried. The coupons were autoclaved at 121 °C for 15 min before the experiments.

### 2.3. Biofilm Formation

All the biofilms were formed on the stainless steel coupons in TSB and chicken juice, according to Papaioannou et al. [8] For mono-species biofilm, 300 μL *L. monocytogenes* or *P. aeruginosa* was added to 3 mL of TSB and chicken juice in 12-well culture plates. The final concentration of *L. monocytogenes* or *P. aeruginosa* with ratios of 1:1 (10^3^ CFU/mL: 10^3^ CFU/mL), 1:10 (10^3^ CFU/mL: 10^4^ CFU/mL), and 10:1 (10^4^ CFU/mL: 10^3^ CFU/mL) were added to 3 mL of TSB and chicken juice in 12-well culture plates. The coupons were incubated at 26 °C under static conditions for 24 h, 48 h, 96 h, 120 h, and 168 h, respectively. TSB and chicken juice were renewed at 120 h.

After incubation at predetermined time intervals, the stainless steel coupons were washed three times with saline solution to remove loosely attached cells. Subsequently, the coupons were vortexed in a 10 mL saline solution with several glass heads to detach the biofilm cells. PALCAM agar medium and King’s B medium were used to count *L. monocytogenes* or *P. aeruginosa* cells using the drop-plating method. The calculation formula for the difference in the biofilm concentration of bacteria follows Equation (1):(1)$$△N=\frac{\log（N_{1}-N_{2}）}{4}$$

Note: ΔN represents the difference in the biofilm concentration of bacteria in different groups (log CFU/cm^2^), and *N*_1_ and *N*_2_ represent the number of bacteria in the biofilm (CFU/cm^2^).

### 2.4. Enumeration of Planktonic Cells

The growth of planktonic cells was counted at the predetermined time intervals. One mL of cell suspension was diluted 10-fold in saline, and the drop-plating method was used in PALCAM agar medium and King’s B medium to enumerate the colonies.

The calculation formula for the difference in the concentration of planktonic bacteria in the suspension follows:(2)$$△N=\log（N_{1}-N_{2}）$$

Note: ΔN represents the difference in the concentration of planktonic bacteria in different suspensions (log CFU/mL), and *N*_1_ and *N*_2_ represent the number of bacteria in the suspension (CFU/mL).

### 2.5. Motility

The soft-agar plate assays were used to evaluate the swimming and swarming motilities, according to Cong et al. and Hidalgo et al., with slight modifications [21,22]. The swimming agar and swarming agar contained 1% tryptone, 0.5% NaCl, 0.25% glucose, 0.6% agar, 2.5% Luria–Bertani, 0.05% glucose, and 0.5% agar (Qingdao Hope Bio-Technology Co., Ltd., Qingdao, China). Ten microliters of *L. monocytogenes* and *P. aeruginosa* were spotted on the plates. The results were recorded after 24 h, 48 h, 96 h, 120 h, and 168 h at 26 °C as the mean diameter of motility ± SD of replicate values from two independent trials.

### 2.6. Scanning Electron Microscopy (SEM)

SEM was used to study the changes in biofilm structure and morphology (composed of either *L. monocytogenes* or *P. aeruginosa* strains) grown in the chicken juice. Herein, the coverslips were carefully washed with PBS three times and fixed with 2.5% glutaraldehyde at 4 °C for 4 h or overnight and then washed three times with PBS and dehydrated with different gradient concentrations of ethanol (10%, 30%, 50%, 70%, 90%, and 100%, respectively) for 15 min. After freeze-drying, the samples were coated with gold film and observed by SEM (Tescan Mira 3 XH, Tescan, Czech Republic).

### 2.7. Statistical Analysis

The mean values were obtained from three independent experiments with duplicate samples. The statistical analysis was conducted by ANOVA, and SPSS software (Statistical Package for the Social Sciences, version 17.0; IBM, Armonk, NY, USA) was used to determine statistical explanations for differences in growth kinetic parameters of *L. monocytogenes* and *P. aeruginosa* among treatment groups. Duncan’s multiple range test and Pearson’s coefficient were applied to compare means. Differences were considered significant if the *p*-value was less than 0.05.

## 3. Results

### 3.1. Formation of Mono and Dual-Species Biofilms

The biofilm formation of *L. monocytogenes* and *P. aeruginosa* are represented in Figure 1 and Figure 2, respectively. In TSB, the *L.* monocytogenes concentration in the dual-species biofilms decreased compared to the mono-species biofilms (3 log CFU/mL) (*p* < 0.05). Compared with mono-species biofilms (3 log CFU/mL), the concentration of *P. aeruginosa* in the dual-species biofilms (1:1) increased by 6.69 and 7.94 log CFU/cm^2^, respectively, at 120 and 168 h, respectively (*p* < 0.05). The concentration of *P. aeruginosa* in the dual-species biofilms (1:10) was increased by 6.42 log CFU/cm^2^ (*p* < 0.05) compared to its concentration in the mono-species biofilms at 120 h. The concentration of *P. aeruginosa* had no significant difference in other mono- and dual-species biofilm groups.

In the chicken juice, there was a difference in the concentration of *L. monocytogenes* and *P. aeruginosa* in the dual-species biofilms. The concentration of *L. monocytogenes* in the dual-species biofilms (1:1) decreased by 2.87 log CFU/cm^2^ (*p* < 0.05) at 24 h compared to its concentration in the mono-species biofilms (3 log CFU/mL). In the dual-species biofilms (1:10), the biofilm concentration of *L. monocytogenes* increased by 2.62–3.80 log CFU/cm^2^ at 96–168 h compared to its concentration in the mono-species biofilms (*p* < 0.05). The concentration of *L. monocytogenes* in the dual-species biofilms (10:1) increased by 3.85 log CFU/cm^2^ at 24 h (*p* < 0.05), while the bacterial concentration decreased by 2.93–4.27 log CFU/cm^2^ at 48–168 h (*p* < 0.05), compared to its concentration in the mono-species biofilms (3 log CFU/mL). However, *P. aeruginosa* concentration in the dual-species biofilms was inhibited by *L. monocytogenes* at 24 h. In comparison, its concentration was increased by *L. monocytogenes* at 96 h (*p* < 0.05). At 24 h, *P. aeruginosa* concentration in the mono-species biofilms was 8.33 log CFU/cm^2^ (Figure 2a) and 8.18 log CFU/cm^2^ (Figure 2b), while *P. aeruginosa* concentration in dual-species biofilms (1:1, 1:10, and 10:1) were 7.29, 7.49, and 7.31 log CFU/cm^2^, respectively. Similarly, at 96 h, the concentration of *P. aeruginosa* in mono-species biofilms was 5.55 log CFU/cm^2^ (Figure 2a), and 6.16 log CFU/cm^2^ (Figure 2b), while *P. aeruginosa* concentration in dual-species biofilms (1:1, 1:10, and 10:1) were 6.54, 7.34, and 7.28 log CFU/cm^2^, respectively. In addition, at 120 h, compared to the mono-species biofilms (Figure 2a) the concentration of *P. aeruginosa* in dual-species biofilms (1:10) significantly increased by 6.21 log CFU/cm^2^ (*p* < 0.05). In contrast, its concentration in the dual-species biofilms (1:1 and 10:1) significantly decreased by 6.04 and 5.88 log CFU/cm^2^ (*p* < 0.05), respectively.

The biofilm formation in TSB and chicken juice varied. The *L. monocytogenes* concentration in the chicken juice (2.73–4.26 log CFU/cm^2^) was significantly lower than those in TSB (3.79–7.92 log CFU/cm^2^) (*p* < 0.05), which means chicken juice had a negative impact on the biofilm formation of *L. monocytogenes* (Figure 1). The different culture mediums also affected the formation of *P. aeruginosa* biofilms (Figure 2). In chicken juice, the concentration of *P. aeruginosa* in the mono- and dual-species biofilms significantly increased at 24 h and 48 h (*p* < 0.05) but significantly decreased at 120 h (*p* < 0.05) compared to its concentration in TSB. At 168 h, the *P. aeruginosa* concentration in the dual-species biofilm (1:1) in the chicken juice was significantly lower than its concentration in TSB (*p* < 0.05). At 96 h, the *P. aeruginosa* concentration in the mono-species biofilms (10^3^ CFU/mL, Figure 2a) in the chicken juice was lower than in TSB. However, its concentration in dual-species biofilm (1:10) in the chicken juice was significantly higher than that in TSB (*p* < 0.05). These results showed that the culture medium significantly influences biofilm formation.

The difference in strains also may lead to a difference in biofilm formation. The maximum concentration of *P. aeruginosa* in TSB and chicken juice were 7.96 and 8.33 log CFU/cm^2^, respectively. The maximum concentration of *L. monocytogenes* in TSB and chicken juices was 6.73 and 4.26 log CFU/cm^2^, respectively. In TSB or chicken juice, the concentration of *P. aeruginosa* was significantly higher than *L. monocytogenes* (*p* < 0.05) in the biofilms.

Incubation time resulted in the dynamic changes in the *L. monocytogenes* and *P. aeruginosa* biofilm formation. In TSB, the biofilm concentration of *L. monocytogenes* significantly increased from 6.86 to 8.09 log CFU/cm^2^ at 168 h to 7.12–8.10 log CFU/cm^2^ at 120 h (*p* < 0.05). In the chicken juice, the concentration of *L. monocytogenes* in the mono- and dual-species biofilms (1:1 and 10:1) gradually decreased with the extension of incubation time. Except for the dual-species biofilms (1:10), the concentration of *P. aeruginosa* in TSB and chicken juice were significantly increased from 7.83 to 8.67 log CFU/cm^2^ at 120 h to 7.93–8.81 log CFU/cm^2^ at 168 h (*p* < 0.05).

### 3.2. Competition in Planktonic Cells

The planktonic cell populations of *L. monocytogenes* and *P. aeruginosa* are shown in Figure 3 and Figure 4, respectively. After the interaction between *L. monocytogenes* and *P. aeruginosa* in TSB and chicken juice, the concentration of planktonic *L. monocytogenes* and *P. aeruginosa* in the dual-species suspension was different from the mono-species suspension. For comparison, *L. monocytogenes* or *P. aeruginosa* in the mono-species biofilms with the initial inoculum concentration of 3 log CFU/mL data was used. In TSB, the planktonic *L. monocytogenes* concentration in the dual-species suspension (1:1) during 48–168 h significantly decreased by 6.65–8.06 log CFU/mL compared to the mono-species suspension (*p* < 0.05). The concentration of planktonic *L. monocytogenes* in the dual-species suspension (1:10) was significantly reduced by 6.32–8.52 log CFU/mL during 24–96 h while increased by 7.82 log CFU/mL at 120 h and significantly decreased by 8.03 log CFU/mL (*p* < 0.05) at 168 h, compared the mono-species suspension. When cultured with *P. aeruginosa* in a 10:1 ratio suspension for 48 h, planktonic *L. monocytogenes* concentration decreased by 6.86 log CFU/mL (*p* < 0.05) compared with those of the mono-species suspension. Compared with the mono-species suspension, after 48 h of incubation, the concentration of planktonic *P. aeruginosa* in the dual-species suspension (1:1, 1:10, and 10:1) was significantly reduced by 7.21, 7.53, and 7.90 log CFU/mL (*p* < 0.05), respectively. In addition, compared with those in the mono-species suspension, the concentration of planktonic *P. aeruginosa* in the dual-species suspension (1:1 and 10:1) decreased by 8.30–8.85 log CFU/mL (*p* < 0.05) after 120 h and 168 h of incubation. In comparison, planktonic *P. aeruginosa* concentration in the dual-species suspension (1:10) significantly increased by 7.81 and 8.24 log CFU/mL at 120 h and 168 h (*p* < 0.05), respectively. The concentration of planktonic *P. aeruginosa* in the dual-species suspension (1:1) was significantly reduced by 7.32 log CFU/mL (*p* < 0.05) after 96 h of incubation, compared with those in the mono-species suspension.

In the chicken juice, compared to the mono-species suspension, the planktonic *L. monocytogenes* concentration in the dual-species suspension (1:1) was significantly reduced by 5.47 and 4.84 log CFU/mL (*p* < 0.05) after 48 h and 96 h, respectively, but was increased by 4.81 log CFU/mL (*p* < 0.05) after 120 h. Compared to the mono-species suspension, the planktonic *L. monocytogenes* concentration in the dual-species suspension of 1:10 significantly decreased by 5.49 and 4.65 log CFU/mL at 24 h and 120 h, respectively, but increased by 5.78 log CFU/mL at 96 h (*p* < 0.05). In the dual-species suspension (10:1), *L. monocytogenes* significantly increased by 4.75–5.98 log CFU/mL (*p* < 0.05) at any time point of incubation compared to the mono-species suspension. The concentration of planktonic *P. aeruginosa* in the dual-species suspension (1:1, 1:10, and 10:1) was significantly inhibited by *L. monocytogenes* (*p* < 0.05) after 24 h and 120 h of incubation (*p* < 0.05). Compared to the mono-species suspension, the planktonic *P. aeruginosa* concentration decreased by 6.33, 7.82, and 6.60 log CFU/mL in the dual-species suspension (1:1, 1:10, and 10:1) after 24 h of incubation, respectively, and decreased by 6.90, 8.75, and 7.89 log CFU/mL after 120 h of incubation, respectively. Compared with the mono-species suspension, the planktonic *P. aeruginosa* concentration in the dual-species suspension (1:1 and 10:1) decreased by 7.76 and 7.78 log CFU/mL after 48 h of incubation, respectively. Compared with mono-species suspension, after 48 h of incubation, the concentration of planktonic *P. aeruginosa* in the dual-species suspension (1:10) significantly increased by 7.59 log CFU/mL. However, with a further extension of the incubation time to 96 h, the *P. aeruginosa* concentration decreased by 8.74 log CFU/mL.

In different culture mediums, the change in planktonic *L. monocytogenes* concentration in the bacterial suspension was similar to the biofilm concentration change in two bacteria. The biofilm formation of *L. monocytogenes* reduced (*p* < 0.05) in chicken juice compared to the TSB medium. The concentration of planktonic *P. aeruginosa* in chicken juice was inhibited at 24 h, while the concentration in chicken juice was increased at 48–96 h (*p* < 0.05) compared to the TSB medium.

In the same culture medium, *L. monocytogenes* and *P. aeruginosa* showed a difference in the planktonic bacterial population. In TSB, the concentration of planktonic *P. aeruginosa* was higher than *L. monocytogenes* in several culture conditions (*p* < 0.05), except for the mono-species suspension (3 log CFU/mL, 48 h; 4 log CFU/mL, 96 h) and the dual-species suspension (1:1, 120 h; 10:1, 48 and 120 h). Similarly, in chicken juice, the concentration of planktonic *P. aeruginosa* (6.87–8.86 log CFU/mL) was higher than those of planktonic *L. monocytogenes* (4.85–6.04 log CFU/mL) (*p* < 0.05).

Incubation time is a factor that leads to a difference in the concentration of planktonic *L. monocytogenes* and *P. aeruginosa*. In TSB, the concentration of planktonic *L. monocytogenes* showed a decreasing trend from 24 to 48 h (*p* < 0.05), except for the dual-species suspension (1:10). In TSB, the concentration of planktonic *L. monocytogenes* showed a decreasing trend from 48 to 96 h (*p* < 0.05), except for the dual-species suspension (10:1). In TSB, the concentration of planktonic *P. aeruginosa* showed a decline from 24 to 96 h (*p* < 0.05) in all the suspensions, except for the dual-species suspension (10:1). In TSB, the concentration of planktonic *P. aeruginosa* was reduced from 24 to 48 h (*p* < 0.05) in the dual-species suspension (1:10). In the chicken juice, the concentration of planktonic *L. monocytogenes* in the mono-species suspension (3 log CFU/mL) and the dual-species suspension (1:10) significantly decreased by 4.88 and 5.74 log CFU/mL, respectively, from 96 to 168 h. In the chicken juice, except for the mono-species suspension group (4 log CFU/mL), the concentration of planktonic *P. aeruginosa* showed a significant increase from 24 to 96 h (*p* < 0.05). From 120 h to 168 h, the concentration of planktonic *P. aeruginosa* showed a downward trend in the mono-species suspension, while its concentration showed an upward trend in the dual-species suspension; among them, the trend was significant (*p* < 0.05) in the mono-species suspension (3 log CFU/mL) and the dual-species suspension (1:1 and 1:10).

### 3.3. Motility

The motility of *L. monocytogenes* and *P. aeruginosa* is shown in Figure 5. The swimming and swarming motility of *L. monocytogenes* and *P. aeruginosa* had similar trends, and the swimming motility increased with the extension of culturing time. At 96, 120, and 168 h, the swimming ability of *P. aeruginosa* was greater than that of *L. monocytogenes* (*p* < 0.05) (Figure 5a). The different initial inoculum levels had specific effects on the swimming motility. The swarming motility of *L. monocytogenes* with the initial inoculum concentration of 3 log CFU/mL at 96, 120, and 168 h was significantly higher than that of *L. monocytogenes* with the initial inoculum level of 4 log CFU/mL (*p* < 0.05). Similar results were observed in the swimming and swarming motility of *P. aeruginosa* at 120 h.

The correlation coefficient between motility and biofilm formation of *L. monocytogenes* and *P. aeruginosa* is shown in Table 1. In TSB, the biofilm formation of *L. monocytogenes* was not correlated with its motility. In contrast, the biofilm formation of *P. aeruginosa* was positively correlated with the swimming motility and swarming motility of *P. aeruginosa* in the dual-species biofilms (1:1) (*p* < 0.05). In the chicken juice, there was no correlation between the biofilm formation and motility of *P. aeruginosa*. It was shown that there was a negative correlation between the biofilm formation and motility of *L. monocytogenes* (*p* < 0.05) in the mono-species biofilms (3 log CFU/mL). The biofilm formation of *L. monocytogenes* was negatively correlated with the swarming motility of *L. monocytogenes* (*p* < 0.05) in the mono-species biofilms (3 log CFU/mL) and the dual-species biofilms (10:1).

### 3.4. Biofilm Structure Observation by SEM

SEM was used to observe the structure of *L. monocytogenes* and *P. aeruginosa* in the mono- and dual-species biofilms cultured in chicken juice for 24 h (Figure 6). *Listeria monocytogenes* in the mono-species biofilms did not form a more prominent structure (Figure 6a), which is in an early adhesion stage. At the same time, *P. aeruginosa* has a relatively dense and stable biofilm structure with EPS (Figure 6b). The structure of two bacteria in the dual-species biofilms was similar to *P. aeruginosa* in the mono-species biofilms. These results showed that *P. aeruginosa* might be dominant in the dual-species biofilm.

## 4. Discussion

In the natural food environment, bacteria mainly exist in dual-species or multispecies biofilms [23,24]. Due to the interaction among bacteria, there are differences in the formation of mono- and dual-species biofilms. Pang et al. compared the mono- and dual-species biofilm of *Salmonella* spp. and *P. aeruginosa* [25], which illustrated that in the mono-species biofilms, *P. aeruginosa* and *Salmonella typhimurium* were reduced than in dual-species biofilms (*p* < 0.05). Most current research focuses on the biofilm formation after culturing at the same concentration [26]. However, bacteria exist in different concentration ratios in a complex microorganism environment than in the mono-species biofilms. This study designed three mixing ratios of 1:1, 1:10, and 10:1 of *L. monocytogenes* and *P. aeruginosa* to match the natural environment. Our results showed that different ratios significantly impacted the formation of *L. monocytogenes* and *P. aeruginosa* biofilms. *Salmonella typhimurium* and *L. monocytogenes* were mixed in five ratios of 1:1, 1:2, 2:1, 1:3, and 3:1 and showed different biofilm formation [27].

Different culture mediums have a different impact on biofilm formation. Compared with the laboratory media, meat juice can affect the morphology and distribution of biofilms [28]. The study compared the biofilm formation of *L. monocytogenes* in TSB and chicken juice, indicating that chicken juice had a significant inhibitory effect on the biofilm formation of *L. monocytogenes*. Liu and Shi also found that the biofilm formation of *L. monocytogenes* in chicken juice was weak [29].

Another study demonstrated that *Pseudomonas* spp. dominated in multispecies biofilms, which could inhibit the population of other species in the biofilms [4]. In this study, *P. aeruginosa* was the dominant bacteria in the dual-species biofilms in TSB, which inhibited the formation of *L. monocytogenes* biofilms. In the chicken juice, *P. aeruginosa* dominated the dual-species biofilms, but it could not inhibit *L. monocytogenes*. Pang et al. demonstrated that *P. aeruginosa* was not dominant in the biofilms formed by *Salmonella enteritidis* and *P. aeruginosa* in the chicken juice at 1:1 and 0.01:1 [30]. However, in TSB, *P. aeruginosa* was the dominant bacteria [6]. Aleksandra et al. [31] found that the population of *P. aeruginosa* was higher than that of *L. innocua* in the two bacterial biofilms, but *P. aeruginosa* was not the dominant bacterium. Different incubation times also could affect the formation of *L. monocytogenes* and *P. aeruginosa* biofilms. Moreno et al. found that there was no significant difference in the biofilms after mixing the five species of bacteria, which contained *L. monocytogenes* and *P. aeruginosa* at 25 °C in TSB or meat juice for 48–180 h (*p* ≥ 0.05) [32]. Our results differed from a previous study. The interaction of the five species in the biofilms is different from that of dual-species biofilms. In addition, the sampling point also could affect the biofilms. The biofilm population of *P. aeruginosa* declined at 48 h and increased at 120 h and 168 h in the chicken juice (*p* < 0.05). One of the possible reasons is that the starvation condition resulted in the decrease in *P. aeruginosa* biofilm concentration [33]. At the same time, renewed culture media at 120 h could support some nutritional components to improve the biofilm formation of *P. aeruginosa*. Moreover, the composition of the media also affected the formation of biofilms. Therefore, the biofilm formation at different incubation times depends on the culture medium’s nutritional conditions.

The interaction between *L. monocytogenes* and *P. aeruginosa* in the bacterial suspension and biofilm was different. In TSB, planktonic *L. monocytogenes* concentration was partially inhibited in the dual-species suspension. In addition, planktonic *L. monocytogenes* was significantly inhibited by chicken juice. The concentration of planktonic *P. aeruginosa* in TSB and chicken juice was higher than that of *L. monocytogenes* partially. These changes in *L. monocytogenes* and *P. aeruginosa* in the suspension were similar to those in biofilms. Wang et al. [5] compared the amount of biofilm formed by different serotypes of *Salmonella* spp. and found that there was no significant correlation between the biofilm formation and growth of *Salmonella*. Studies have shown that biofilm formation also depends on the adhesion ability of the bacteria, so the planktonic bacteria in the bacterial suspension could not be used for the evaluation of bacterial biofilm formation [34].

This study showed no significant correlation between biofilm formation and motility. Bonaventura et al. found no correlation between biofilm formation and motility in 44 strains of *L. monocytogenes* at 4 °C, 12 °C, and 22 °C [35]. Studies have shown that the swarming motility of *P. aeruginosa* is related to nutritional conditions [36]. The swimming motility of bacteria, which is the characteristic responsible for the movement of flagella, may be helpful in the adhesion and formation of initial biofilms. However, biofilm formation still depends on the culture conditions [37].

Our study of biofilm in the chicken juice incubated for 24 h showed that the mono-species biofilms of *L. monocytogenes* are weaker than *P. aeruginosa* biofilms. The reasons for this condition are mentioned as follows: First, chicken juice could inhibit the growth and biofilm formation of *L. monocytogenes*. Therefore, *L. monocytogenes* was loosely attached to the coupons at 24 h. Second, the initial inoculum concentration of *L. monocytogenes* was 3 or 4 log CFU/mL, which might be too low to form the biofilm than the high initial inoculum concentration [8]. Third, *P. aeruginosa* could produce more EPS and was a stronger biofilm-producer than *L. monocytogenes* [38]. In the dual-species biofilms, the structure observed by SEM is similar to the mono-species biofilms of *P. aeruginosa*. *Pseudomonas aeruginosa* showed a dominant status and protected the dual species from disinfectants and environmental stress [39].

Although some tryptone (TSB) media are rich in nitrogen and amino acids and can be used as good microbiological media, the complexity of real food matrices (e.g., different nutrient availability and pH) makes it difficult to represent the real situation in the food system. Therefore, some meat juice was used as an ideal culture medium for bacterial biofilm to simulate the growing condition in a meat processing environment. In our study, chicken juice (exudate from frozen chicken breast) was used instead of growth medium to simulate the real situation in a meat processing environment and to better understand the relationship between the chicken environment and pathogenic bacteria. The results will help promote the safety of chicken processing as it mimics the composition and microbiome of chicken for more predictable results in polluted environments. However, considering that our research is performed based on the laboratory, the real environment is often much more complicated than that of the laboratory, and a single treatment cannot completely simulate the complex background of a food processing plant. Therefore, it is necessary to further consider the influence of various factors (such as microbial population, pH, and nutrient composition) to be able to resemble the real food environment more closely.

## 5. Conclusions

This article aimed to evaluate the biofilm formation, planktonic bacterial concentration, and motility of *L. monocytogenes* and *P. aeruginosa* in mono- and dual-species biofilms under a simulated chicken processing environment. The results showed that the biofilm formation of *L. monocytogenes* and *P. aeruginosa* was significantly affected by multiple factors, including the interaction among bacteria, culture medium conditions, the ratio of two different kinds of bacteria, and incubation time. The chicken juice had an inhibitory effect on *L. monocytogenes* (*p* < 0.05). In TSB, *P. aeruginosa* was dominant in the biofilms. It is difficult to describe the dynamic process of biofilm formation with a particular trend because the biofilm formation mainly depends on the culture medium conditions. The growth of planktonic *L. monocytogenes* and *P. aeruginosa* in the suspension differed from those in the biofilms. Thus, the change in planktonic bacterial concentration cannot be used for judging bacterial biofilm formation. Finally, the motility of *L. monocytogenes* and *P. aeruginosa* had no significant correlation with the biofilm formation.

This study helps understand the biofilm formation of *L. monocytogenes* and *P. aeruginosa* in natural foods and provides the theoretical basis for the prevention and control of *L. monocytogenes* and *P. aeruginosa* biofilms in the future.

## Figures and Tables

**Figure 1 foods-11-01917-f001:**
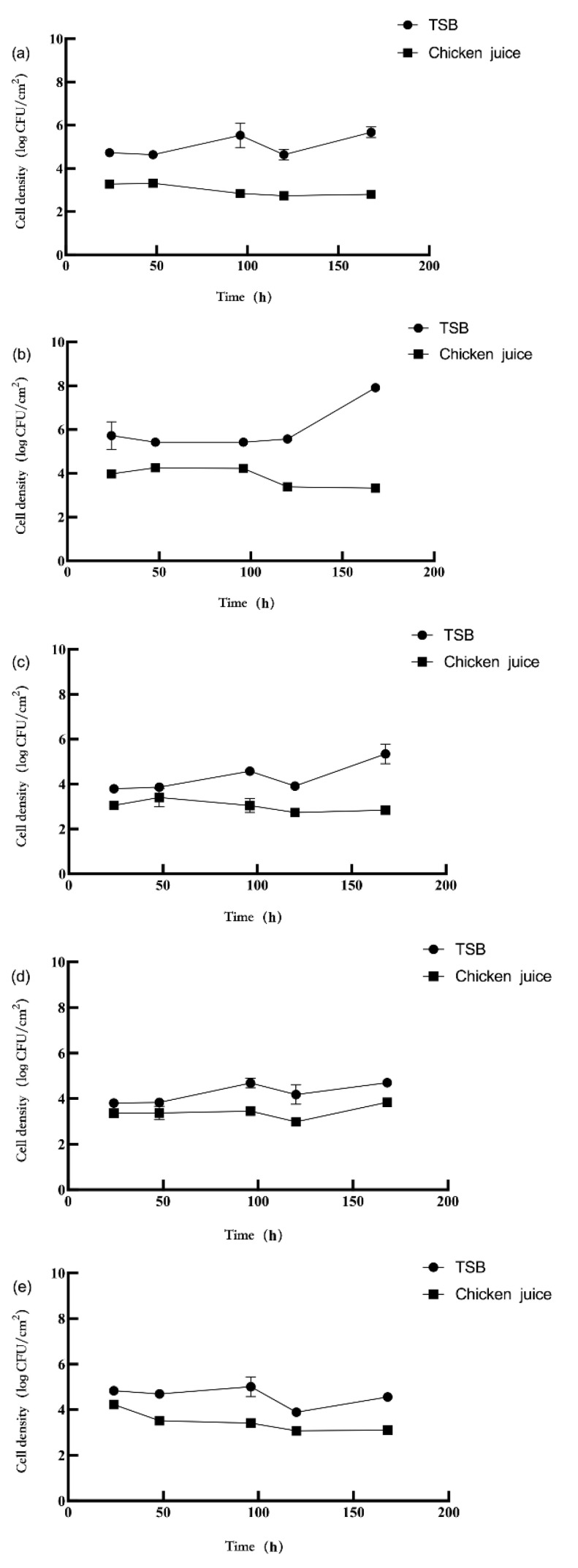
Biofilm formation of *L. monocytogenes* (**a**–**e**) in different incubation conditions. Note: The mono-species biofilms of *L. monocytogenes* with the initial inoculum concentration of 3 log CFU/mL (**a**) and 4 log CFU/mL (**b**); the dual-species biofilms of *L. monocytogenes* and *P. aeruginosa* with the ratio 1:1 (**c**); 1:10 (**d**); and 10:1 (**e**).

**Figure 2 foods-11-01917-f002:**
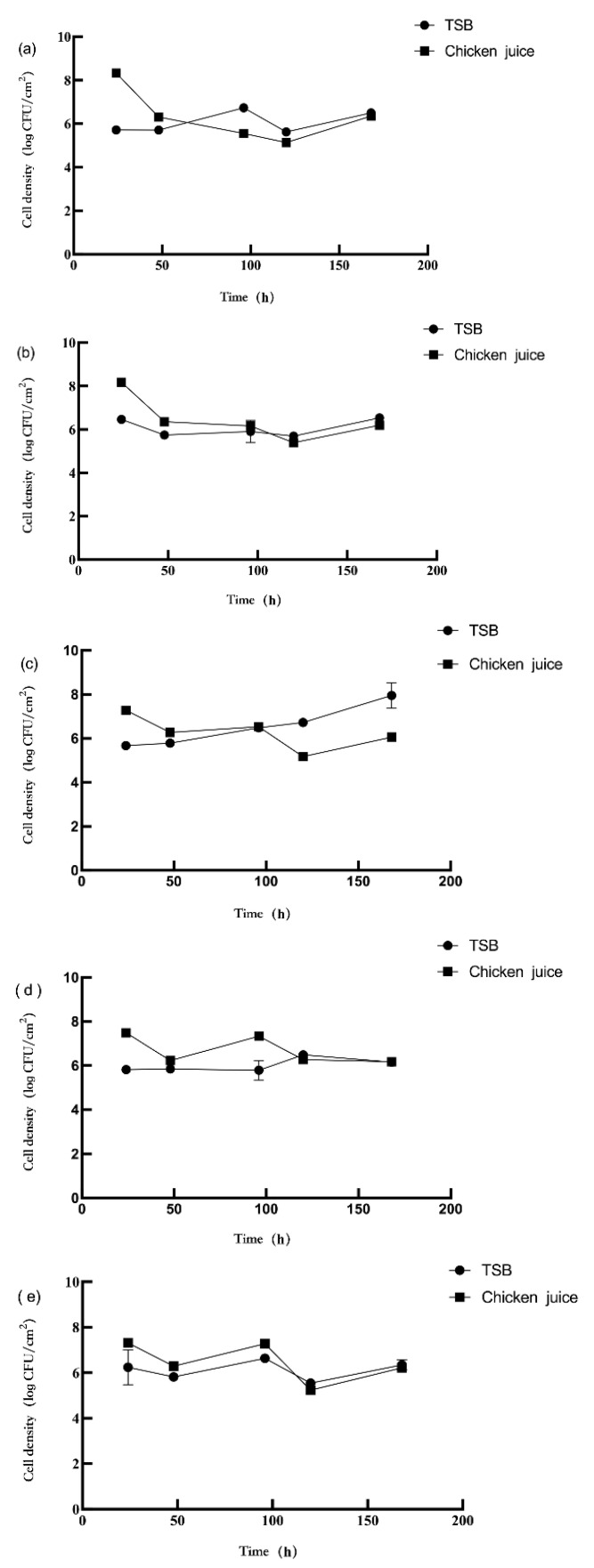
Biofilm formation of *P. aeruginosa* (**a**–**e**) in different incubation conditions. Note: The mono-species biofilms of *P. aeruginosa* with the initial inoculum concentration of 3 log CFU/mL (**a**) and 4 log CFU/mL (**b**); the dual-species biofilms of *L. monocytogenes* and *P. aeruginosa* with the ratio of 1:1 (**c**); 1:10 (**d**); and 10:1 (**e**).

**Figure 3 foods-11-01917-f003:**
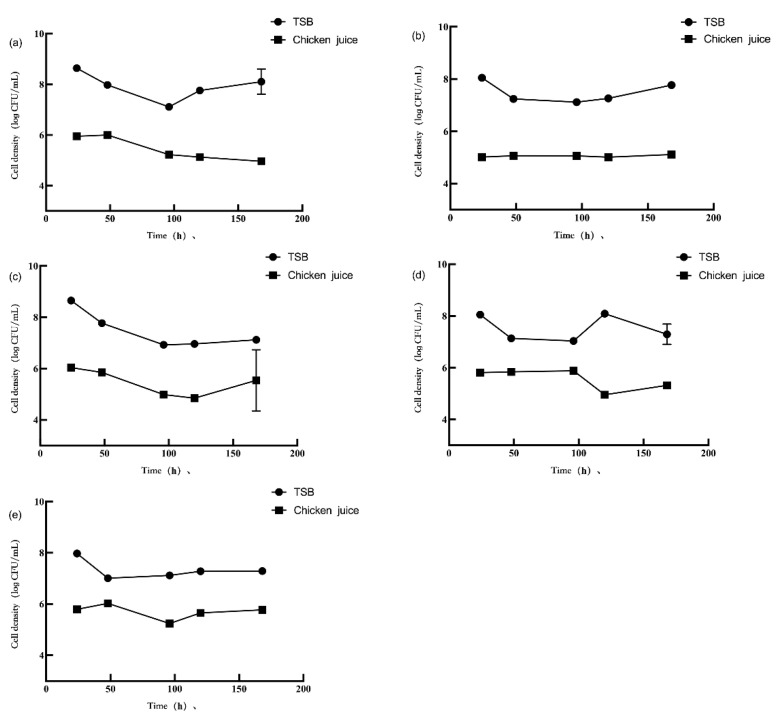
The planktonic population of *L. monocytogenes* (**a**–**e**) in different incubation conditions. Note: The mono-species suspension of *L. monocytogenes* biofilms with the initial inoculum concentration of 3 log CFU/mL (**a**) and 4 log CFU/mL (**b**); the dual-species suspension of *L. monocytogenes* and *P. aeruginosa* biofilms with the ratio 1:1 (**c**); 1:10 (**d**); and 10:1 (**e**).

**Figure 4 foods-11-01917-f004:**
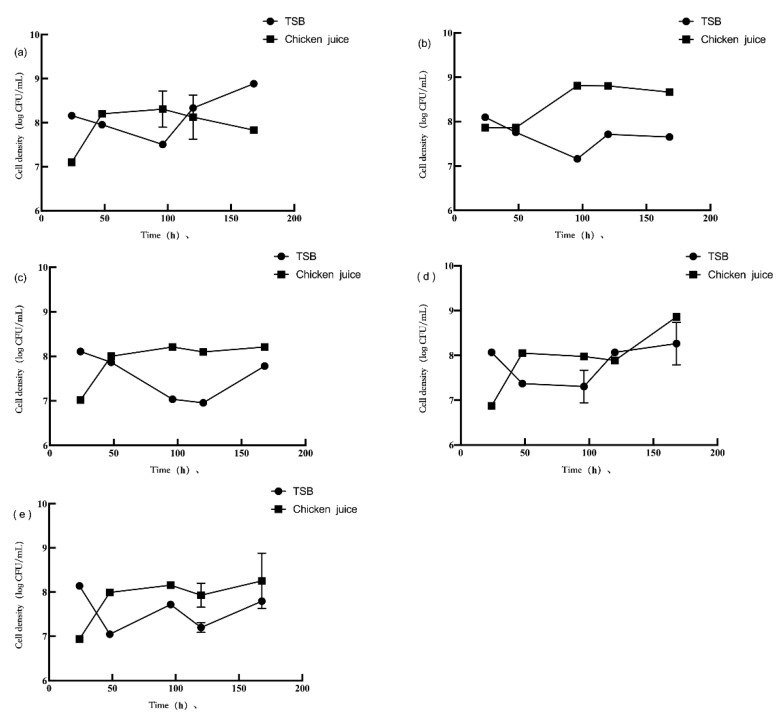
The planktonic population of *P. aeruginosa* (**a**–**e**) in different incubation conditions. Note: The mono-species suspension of *P. aeruginosa* biofilms with the initial inoculum concentration of 3 log CFU/mL (**a**) and 4 log CFU/mL (**b**); the dual-species suspension of *L. monocytogenes* and *P. aeruginosa* biofilms with the ratio 1:1 (**c**); 1:10 (**d**); and 10:1 (**e**).

**Figure 5 foods-11-01917-f005:**
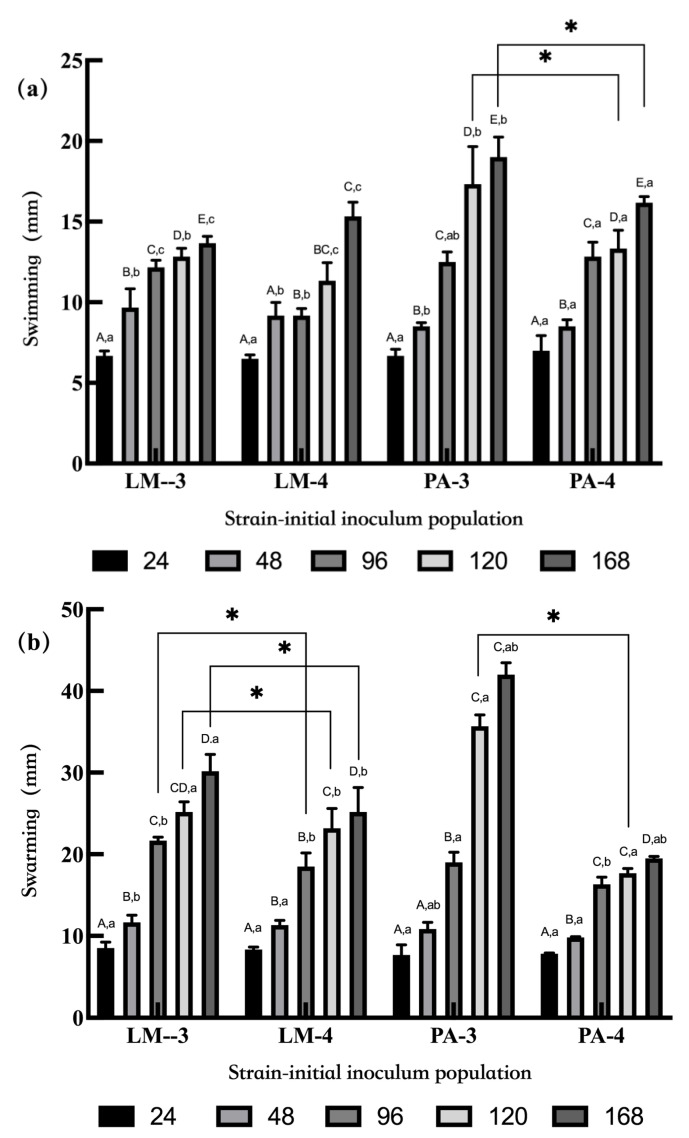
The motility of *L. monocytogenes* and *P. aeruginosa* in different populations and incubation times. Note: (**a**) swimming ability and (**b**) swarming ability. Capital letters (A–D) indicate a significant difference (*p* < 0.05) within different incubation times in the same group. Lowercase letters (a–c) indicate a significant difference (*p* < 0.05) within the same incubation time between LM-3 and PA-3 or LM-4 and PA-4 groups. The asterisks indicate a significant difference (*p* < 0.05) between LM-3 and LM-4 or PA-3 and PA-4 groups.

**Figure 6 foods-11-01917-f006:**
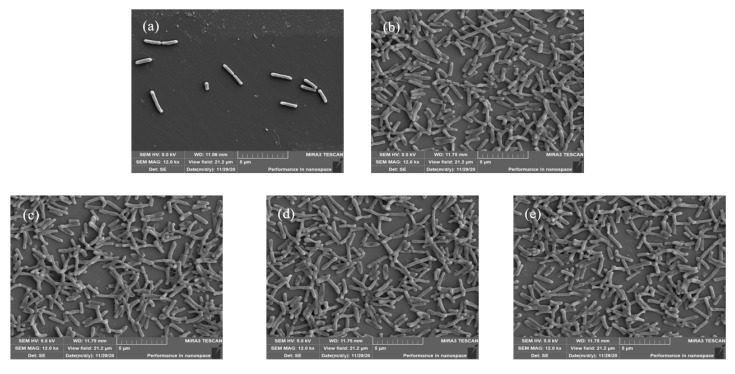
Scanning electron micrographs of *L. monocytogenes* and *P. aeruginosa* biofilms in mono- and dual-species biofilms in the chicken juice at 24 h. Note: The mono-species biofilms of *L. monocytogenes* (**a**) and *P. aeruginosa* (**b**) with the initial inoculum concentration of 3 log CFU/mL; dual-species biofilms of *L. monocytogenes* and *P. aeruginosa* with the ratio of 1:1 (**c**); 1:10 (**d**); and 10:1 (**e**).

**Table 1 foods-11-01917-t001:** The correlation coefficient between motility and biofilm formation of *P. aeruginosa* and *L. monocytogenes* at 26 °C.

Culture Medium	TSB					Chicken Juice				
Ratios Motility	3	4	1:1	1:10	10:1	3	4	1:1	1:10	10:1
*P. aeruginosa*										
Swarming-3	0.315	0.103	0.934 *	0.803	−0.122	−0.564	−0.677	−0.745	−0.615	−0.655
Swimming-3	0.568	−0.038	0.901 *	0.634	0.096	−0.731	−0.770	−0.693	−0.442	−0.479
Swarming-4	0.380	0.019	0.925 *	0.778	−0.086	−0.653	−0.744	−0.767	−0.586	−0.635
Swimming-4	0.616	0.075	0.950 *	0.580	0.163	−0.656	−0.711	−0.629	−0.464	−0.432
*L. monocytogenes*										
Swarming-3	−0.134	0.620	0.769	0.842	−0.445	−0.934 *	−0.729	−0.726	0.279	−0.881 *
Swimming-3	−0.031	0.458	0.689	0.811	−0.429	−0.885 *	−0.565	−0.549	0.198	−0.964 **
Swarming-4	−0.160	0.566	0.696	0.788	−0.536	−0.942 *	−0.756	−0.747	0.179	−0.906 *
Swimming-4	−0.484	0.817	0.801	0.664	−0.429	−0.685	−0.755	−0.524	0.470	−0.832

Note: * means difference (*p* < 0.05); ** means significant differecne (*p* < 0.01).

## Data Availability

Data is contained within the article, more details are available from the corresponding author.

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
