# Peer review of "Biofilm Formation of Listeria monocytogenes and Pseudomonas aeruginosa in a Simulated Chicken Processing Environment"

_foods, 2022, doi:10.3390/foods11131917_

Round 1

Reviewer 1 Report

The Manuscript entitled “Biofilm formation of Listeria monocytogenes and Pseudomonas aeruginosa in a simulated chicken processing environment” addresses an important theme regarding L. monocytogenes and P. aeruginosa biofilms formed as mono- and dual-species biofilms.

First of all, I suggest linguistic proofreading. Sometimes grammatical errors make the text difficult to understand.

It seems to me that in the Abstract one can omit the enumerations (first, second, etc.) and present it in a more informative way.

The Introduction outlines the problems of work very well. The aim is clearly formulated. I suggest minor changes as follow:

Line 49. It will be good to incorporate reference here.

Line 55: multi-bacteria? In my opinion it is bad word

Line 66: Are the Authors sure that there are no newer, more up-to-date reports from EFSA and ECDC?

Line 77: Remove the name in brackets (reference), leave the number

Line 87: L. monocytogenes – italics

The Materials and methods are generally well described. Minor suggestions below:

Line 115-120: Isn't that a repetition? Do I understand correctly? Please explain.

Line 131: enumerate colonies? Colonies were not the aim to evaluate. This value correspond with CFU per… Please detail

Line 142: was used (English correction suggested, mentioned above)

Line 141: SEM poorly described, please describe in more detail

The Results are described too extensively, especially the section on biofilms (mono- and dual) as well as the section on planktonic cells

Line 157: respectively

The description of the results of mono- and dual-biofilm formation is too detailed

Line 221 and 229: “in different incubation conditions” – it is too general, please specify

Line 233: respectively

The descriptions of the axes in Figures 1-4 are too small

Competition in planktonic cells – also to long and too detail

Sometimes Authors write h sometimes hours, please standardize the whole Manuscript

Figure 6 – increase the font

The Discussion is written appropriately. The Authors raise every important issue, supporting it with literature.

Please pay attention to citations, references - eg.:

Line 399. Reference no 26 – Should be Kocot et al. not Aleksandra

Line 448: This article – better: our study or our results. Emphasize the importance of Your work/results, and not some article ;)

In general, I believe that the work has been prepared – planned, performed and written in an appropriate manner, in accordance with the standards of scientific publications. My comments are mainly illustrative. If the Authors do not agree with them – please provide a constructive comment or explanation :) The work requires minor corrections before publishing in Foods.

Author Response

Dear,

    Thank you for your review of our manuscript. We appreciate the suggestions and advices. Our point-to-point responses are provided below, text that has been added or modified from the original text is shown in the revised manuscript in yellow font.    

    Upon review of our revised manuscript, we hope that you will find it acceptable for publication and we look forward to your response.

Sincerely

The Manuscript entitled “Biofilm formation of Listeria monocytogenes and Pseudomonas aeruginosa in a simulated chicken processing environment” addresses an important theme regarding L. monocytogenes and P. aeruginosa biofilms formed as mono- and dual-species biofilms.

  1. First of all, I suggest linguistic proofreading. Sometimes grammatical errors make the text difficult to understand.

RE: The manuscript has been checked carefully, text that has been added or modified from the original text is shown in the revised manuscript in yellow font.    

  1. It seems to me that in the Abstract one can omit the enumerations (first, second, etc.) and present it in a more informative way.

RE: The abstract has been revised carefully.

The Introduction outlines the problems of work very well. The aim is clearly formulated. I suggest minor changes as follow:

3.Line 49. It will be good to incorporate reference here.

RE: The reference has been added.

4.Line 55: multi-bacteria? In my opinion it is bad word

RE: “multi-species strain” has been used.

5.Line 66: Are the Authors sure that there are no newer, more up-to-date reports from EFSA and ECDC?

RE:

6.Line 77: Remove the name in brackets (reference), leave the number

RE: It has been changed.

7.Line 87: L. monocytogenes – italics

RE: It has been changed.

The Materials and methods are generally well described. Minor suggestions below:

8.Line 115-120: Isn't that a repetition? Do I understand correctly? Please explain.

RE: It is different. First, treatment of coupons has been described. The coupons can not be used directly without treating using Ethanol and autoclave. Then, the treated coupons are used in experiment.

9.Line 131: enumerate colonies? Colonies were not the aim to evaluate. This value correspond with CFU per… Please detail

RE: Calculation formula for the difference in the concentration of planktonic bacteria has been added.

 Note: means the difference in the concentration of planktonic bacteria in different suspension (log CFU/mL), and  means the amount of bacteria in the suspension (CFU/mL).

  1. Line 142: was used (English correction suggested, mentioned above)

RE:It has been reivsed.

  1. Line 141: SEM poorly described, please describe in more detail

RE: It has been revised.

The Results are described too extensively, especially the section on biofilms (mono- and dual) as well as the section on planktonic cells

  1. Line 157: respectively

RE: It has been added.

The description of the results of mono- and dual-biofilm formation is too detailed

13.Line 221 and 229: “in different incubation conditions” – it is too general, please specify

RE: It has been specified

14.Line 233: respectively

RE: It has been added.

15.The descriptions of the axes in Figures 1-4 are too small

RE: The descriptions of the four figures have been added.

16.Competition in planktonic cells – also to long and too detail

RE: It has been revised.

17.Sometimes Authors write h sometimes hours, please standardize the whole Manuscript

RE: It has been changed as “h”.

18.Figure 6 – increase the font

RE: The figure has been revised.

The Discussion is written appropriately. The Authors raise every important issue, supporting it with literature.

19.Please pay attention to citations, references - eg.:

RE: They have been revised.

20.Line 399. Reference no 26 – Should be Kocot et al. not Aleksandra

RE: It has been revised. And the reference has been added.

21.Line 448: This article – better: our study or our results. Emphasize the importance of Your work/results, and not some article ;)

RE: “Our study of biofilm in chicken juice incubated at 24 h......”

Reviewer 2 Report

The manuscript describes results of Listeria monocytogenes and Pseudomonas aeruginosa in mono- and dual-species. Even though title addresses simulated chicken processing environment' used incubation temperature was 26 °C which is rather high and is not simulating these conditions.

The manuscript needs extensive editing and correction of grammar, language.

Does chicken juice really work inhibitory?  In my opinion, this should be reserved for antimicrobial or anti-biofilm compounds. For such instance, prepared growth media have optimal composition for bacterial growth but chicken juice does not. So growth or biofilm formation is slower, but it does not mean that chicken juice inhibits growth / biofilm formation. This should be corrected throughout the manuscript.

In introduction only one reference (15) is given for known data on L. monocytogenes and P. aeruginosa dual-species biofilm. However, there are more articles available and this should be re-examined in more detail and included in the article.

Is P. aeruginosa really typical chicken meat spoiler? Many other Pseudomonas spp. are reported as important spoilers of chicken meat and P. aeruginosa is not among them.

Results under point 3.1 are giving same numbers in graphs and text. This is not necessary. Results are described in very long way but many things are missed. It is not clear what was compared. It should be clear from graphs what is compared and what is significantly different. Also graphs should be done in such way that sd values are seen. Description of figure should be done with all information. Also under other R subtitles should be revised. 

In my opinion, all report on results could be given in a shorter and more concise version and more emphasis should be placed on the discussion, which is currently very deficient. Many articles are cited but their link to presented results in this manuscript is not given.

L32-33, L88-90 and L465-466: What is this 'theoretical reference for prevention and control of biofilm'? What can be suggested based on presented results? Just stating this is not enough.

L66: there is newer EFSA and ECDC report available and should be used.

L129: From where planktonic cells were counted? From wells where SS coupons were incubated?

L133: What was used as inoculum?

L141: Which biofilm samples were used?

L148: How data normality and homogeneity of variance were tested?

L235: Figure 2?

L322: 168 h - in MM 196 h.

L366: Where are controls for both bacteria if only two figures are given? In dual-species biofilm mark which bacteria is which.

L384: How it induces morphologies?

Author Response

Dear,

    Thank you for your review of our manuscript. We appreciate the suggestions and advices. Our point-to-point responses are provided below, text that has been added or modified from the original text is shown in the revised manuscript in yellow font.    

    Upon review of our revised manuscript, we hope that you will find it acceptable for publication and we look forward to your response.

Sincerely

The manuscript describes results of Listeria monocytogenes and Pseudomonas aeruginosa in mono- and dual-species. Even though title addresses simulated chicken processing environment' used incubation temperature was 26 °C which is rather high and is not simulating these conditions.

The manuscript needs extensive editing and correction of grammar, language.

  1. Does chicken juice really work inhibitory?  In my opinion, this should be reserved for antimicrobial or anti-biofilm compounds. For such instance, prepared growth media have optimal composition for bacterial growth but chicken juice does not. So growth or biofilm formation is slower, but it does not mean that chicken juice inhibits growth / biofilm formation. This should be corrected throughout the manuscript.

RE: Thanks for your comments. In this study, we used a control group (TSB) and an experimental group (chicken juice) to investigate the interaction in biofilm formation by L. monocytogenes and P. aeruginosa. Our results indicated (Fig. 1) that the presence of chicken juice might have a certain influence on the biofilm formation of the two bacteria. First, regarding the nutritional content, studies have found that L. monocytogenes could survive in chicken juice, which means it can provide enough nutrients for the pathogen (Liu & Shi, 2018). However, further studies found that when cultured in different gravies (chicken, pork, and beef), there were significant changes in biofilm formation (Liu & Shi, 2018). The number of biofilms produced by L. monocytogenes in chicken juice was significantly lower than that in pork juice, which may be due to the characteristics of the medium. For example, studies have found that chicken juice contains more creatine, creatinine and free amino acids (Elbir & Oz, 2021). The presence of these components, especially creatinine, is able to affect bacterial replication and thus may lead to differences between biofilms (McDonald et al., 2012).

Reference

Elbir, Z., & Oz, F. (2021). Determination of creatine, creatinine, free amino acid and heterocyclic aromatic amine contents of plain beef and chicken juices. Journal of food science and technology, 58(9), 3293-3302. https://doi.org/10.1007/s13197-020-04875-8 

Liu, A., & Shi, C. (2018). Pork juice promotes biofilm formation in Listeria monocytogenes [https://doi.org/10.1111/jfs.12439]. Journal of Food Safety, 38(2), e12439. https://doi.org/https://doi.org/10.1111/jfs.12439 

McDonald, T., Drescher, K. M., Weber, A., & Tracy, S. (2012). Creatinine inhibits bacterial replication. The Journal of Antibiotics, 65(3), 153-156. https://doi.org/10.1038/ja.2011.131 

2.In introduction only one reference (15) is given for known data on L. monocytogenes and P. aeruginosa dual-species biofilm. However, there are more articles available and this should be re-examined in more detail and included in the article.

RE: The references has been added.

3.Is P. aeruginosa really typical chicken meat spoiler? Many other Pseudomonas spp. are reported as important spoilers of chicken meat and P. aeruginosa is not among them.

RE: Thanks for your comments. Through literature search, we found that Pseudomonas spp. is one of the common spoilage bacteria in meat products, while P. aeruginosa is detected in chicken meat. Considering that P. aeruginosa is also a conditional pathogen, we think that the research has some significance.

4.Results under point 3.1 are giving same numbers in graphs and text. This is not necessary. Results are described in very long way but many things are missed. It is not clear what was compared. It should be clear from graphs what is compared and what is significantly different. Also graphs should be done in such way that sd values are seen. Description of figure should be done with all information. Also under other R subtitles should be revised. 

RE:They have been revised.

5.In my opinion, all report on results could be given in a shorter and more concise version and more emphasis should be placed on the discussion, which is currently very deficient. Many articles are cited but their link to presented results in this manuscript is not given.

RE: They have been revised.

6.L32-33, L88-90 and L465-466: What is this 'theoretical reference for prevention and control of biofilm'? What can be suggested based on presented results? Just stating this is not enough.

RE: The experimental results show that biofilm formation in the real environment simulation medium is different from the one in the laboratory medium, so the research on the prevention and control of the biofilm can use the real environment simulation medium, which is more realistic.

  1. L66: there is newer EFSA and ECDC report available and should be used.

RE: CDC data has been updated.

8.L129: From where planktonic cells were counted? From wells where SS coupons were incubated?

RE: Thanks for your comments. We created the same culture environment as well as the biofilm in which planktonic cells were cultured to ensure consistent results.

9.L133: What was used as inoculum?

RE: TSBYE has been used as inoculum in this study.

10.L141: Which biofilm samples were used?

11.L148: How data normality and homogeneity of variance were tested?

RE:  In the statistical analysis by SPSS, the normality and homogeneity of variance were tested.

12.L235: Figure 2?

RE: It has been revised.

13.L322: 168 h - in MM 196 h.

RE: In MM, it has been revised as 168h.

Round 2

Reviewer 2 Report

The manuscript was improved to some point but not all comments were addressed. In graphs, statistics data is still not presented so figures are not self-explanatory. Please also review what is given as last name in references - in reference 19 first names are given.

L65: Reference 12 is EFSA and ECDC and not CDC. The newest report should be used. Please correct with data from newest report available here: https://www.efsa.europa.eu/en/efsajournal/pub/6971

L80: These references should be used to improve discussion. Some more references on the topic - review full articles, not only abstracts:

https://www.frontiersin.org/articles/10.3389/fmicb.2018.01706/full

https://www.frontiersin.org/articles/10.3389/fmicb.2016.00134/full

https://pubmed.ncbi.nlm.nih.gov/31113117/

L155: Bacterial names in italic.

L161: Add which tests were used to test normality and homogeneity of variance.

Author Response

Dear,

  Thank you for your review of our manuscript. We appreciate the suggestions and advices. Our point-to-point responses are provided below, text that has been added or modified from the original text is shown in the revised manuscript in yellow font.    

    Upon review of our revised manuscript, we hope that you will find it acceptable for publication and we look forward to your response.

Comments:

*Does the "chicken juice" presented in this manuscript really correspond to a "real environment"? Some tryptones (TSB medium) can be made from enzymatic digestions of meats, one can wonder if in this case it does not correspond more to real conditions than sterile distilled water homogenized (stomacher?) with chicken pectoral muscles. It is not very clear in the manuscript whether the meat is presented with or without the skin? This is a real difference that needs to be discussed. The qualities and shortcomings of the "real environment", as well as its limitations, need to be presented more clearly in this article and discussed in more detail.

RE: Thanks for your comments. In accordance with Reviewer’s comment, we have added a discussion about "chicken juice" to the revised manuscript as follows (line 481-496):

Although some tryptone (TSB) media are rich in nitrogen and amino acids and can be used as good microbiological media, the complexity of real food matrices (e.g., different nutrient availability and pH) makes it difficult to represent the real situation in the food system [40]. Therefore, some meat juice was used as an ideal culture medium for bacterial biofilm to simulate the growing condition in meat processing environment. In our study, chicken juice (exudate from frozen chicken breast) was used instead of growth medium to simulate the real situation in a meat processing environment and to better understand the relationship between the chicken environment and pathogenic bacteria.

The results will help promote the safety of chicken processing as it mimics the composition and microbiome of chicken for more predictable results in polluted environments. However, considering that our research is carried out based on the laboratory, the real environment is often much more complicated than that of the laboratory, and a single treatment cannot completely simulate the complex background of a food processing plant. Therefore, it is necessary to further consider the influence of various factors (such as microbial population, pH, and nutrient composition) to be able to resemble the real food environment more closely.

Comments and Suggestions for Authors

The manuscript was improved to some point but not all comments were addressed. In graphs, statistics data is still not presented so figures are not self-explanatory.

  1. Please also review what is given as last name in references - in reference 19 first names are given.

RE: It has been revised.

L65: Reference 12 is EFSA and ECDC and not CDC. The newest report should be used. Please correct with data from newest report available here: https://www.efsa.europa.eu/en/efsajournal/pub/6971

RE: It has been corrected.

L80: These references should be used to improve discussion. Some more references on the topic - review full articles, not only abstracts:

https://www.frontiersin.org/articles/10.3389/fmicb.2018.01706/full

https://www.frontiersin.org/articles/10.3389/fmicb.2016.00134/full

https://pubmed.ncbi.nlm.nih.gov/31113117/

RE: It has been revised.

L155: Bacterial names in italic.

RE: It has been revised.

L161: Add which tests were used to test normality and homogeneity of variance.

RE: The mean values were obtained from three independent experiments with duplicate samples. The statistical analysis was conducted by ANOVA, and SPSS software (Statistical Package for the Social Sciences, version 17.0; IBM, NY, USA) was used to determine statistical explanations for differences in growth kinetic parameters of L. monocytogenes and P. aeruginosa between treatment groups, and Duncan's multiple range test and Pearson's coefficient were applied to compare means. Differences were considered significant if the p-value was less than 0.05.
